# Phylogenetic Analysis of Spliceosome SF3a2 in Different Plant Species

**DOI:** 10.3390/ijms24065232

**Published:** 2023-03-09

**Authors:** Yuan Tian, Debatosh Das, Min Li, Tao Song, Jingfang Yang, Yinggao Liu

**Affiliations:** 1State Key Laboratory of Crop Biology, College of Life Science, Shandong Agricultural University, Taian 271018, China; 2Co-Innovation Center for Sustainable Forestry in Southern China, College of Biology and the Environment, Nanjing Forestry University, Nanjing 210037, China; 3College of Agriculture, Food and Natural Resources, Division of Plant Sciences and Technology, 52 Agricultural Building, University of Missouri-Columbia, Columbia, MO 65201, USA; 4State Key Laboratory for Biology of Plant Diseases and Insect Pests, Institute of Plant Protection, Chinese Academy of Agricultural Sciences, Beijing 100193, China

**Keywords:** alternative splicing, spliceosome, *SF3a2*, gene family, cladogram, plants

## Abstract

The formation of mature mRNA requires cutting introns and splicing exons. The occurrence of splicing involves the participation of the spliceosome. Common spliceosomes mainly include five snRNPs: U1, U2, U4/U6, and U5. *SF3a2*, an essential component of spliceosome U2 snRNP, participates in splicing a series of genes. There is no definition of *SF3a2* in plants. The paper elaborated on *SF3a2s* from a series of plants through protein sequence similarity. We constructed the evolutionary relationship of *SF3a2s* in plants. Moreover, we analyzed the similarities and differences in gene structure, protein structure, the cis-element of the promoter, and expression pattern; we predicted their interacting proteins and constructed their collinearity. We have preliminarily analyzed *SF3a2s* in plants and clarified the evolutionary relationship between different species; these studies can better serve for in-depth research on the members of the spliceosome in plants.

## 1. Introduction

As we all know, in eukaryotes, the coding regions of genes are discontinuous in higher organisms; there are coding exons and noncoding introns [1,2]. The excision of noncoding introns and the splicing of coding exons through the spliceosome is called splicing, a necessary process for the formation of mature mRNA. Introns are defined by three important sites, the 5′ splice site (5′SS), branch point (BP) adenosine, and 3′ splice site (3′SS); the three sites all have short conserved sequences [3]. The occurrence of splicing requires the participation of the spliceosome. Introns can be divided into two categories: U2-dependent introns and U12-dependent introns; the splicing of U2-dependent introns requires U1 snRNP, U2 snRNP, U4/U6 snRNP, and U5 snRNP. U1 snRNA of the U1 snRNP complex recognizes the 5′SS splice site of the intron and U2 snRNA of the U2 snRNP complex recognizes the branch site and 3′SS splice site of the intron; then, the U4/U5/U6 snRNP complexes were recruited, the intron was excised after forming a lasso structure, and each complex dissociated to prepare for the subsequent reaction [4,5].

The U12-type introns are removed by the secondary spliceosome, which contains five snRNPs: U11, U12, U4atac, U5, and U6atac. U5 snRNP is shared between the two spliceosomes. The U2-type intron motifs are shorter and more variable, while the U12-type introns usually have more prolonged and more conserved sequence motifs. Despite differences at the sequence level, similar snRNA in the two spliceosomes fold into similar secondary structures [6,7,8].

Alternative splicing is an essential gene regulatory process that generates multiple transcripts from a single gene. It increases the diversity of transcriptome and proteome; it mainly includes the following five different types: (1) intron retention (IR); (2) exon skipping (ES); (3) mutually exclusive exons (MEEs); (4) alternative 5′ splice sites (5′SS); and (5) alternative 3′ splice sites (3′SS) [3,9,10]. It is reported that more than 95% of pre-mRNAs are alternatively spliced in mammals. The dysregulation of alternative splicing is implicated in many diseases, including neurological disorders, cardiovascular and metabolic diseases, and cancers [11,12]. In plants, more than 70% of genes undergo alternative splicing events, participate in plant growth and development, and participate in various abiotic stresses, such as high-temperature, low-temperature, salt stress, drought stress, and abscisic acid (ABA) [13,14,15,16,17].

A spliceosome is an essential component produced by splicing. Except for typical U1 snRNP, U2 snRNP, U4/U6 snRNP, and U5 snRNP, the spliceosome also contains the following three important components: nineteen complexes (NTCs), the NTC-related complex (NTR), and the retention and splicing complex (RES) [4,18,19,20]. U2 snRNP is the main complex of intron 3′ splice site recognition; U2 snRNP mainly includes the SF3a complex (three proteins: SF3a120, SF3a66, and SF3a60), SF3b complex (seven proteins: SF3b155, SF3b145, SF3b130, SF3b49, SF3b10, SF3b14a, and SF3b14b), U2 snRNA, and nine proteins (U2-A′, U2-B′′, and the heptameric U2 Sm complex) interacting with U2 snRNA [4,21,22,23]. The three proteins of the SF3a complex exhibit an extended conformation, and SF3a60 bridges the gap between SF3a120 and SF3a66. A 27-amino-acid region mediates the SF3a66-SF3a120 interaction in the SF3a120 C-terminal to the second suppressor-of-white-apricot and prp21/spp91 domain and amino acids 108–210 of SF3a66 [24]. In addition, SF3a66 (SF3a2) is also involved in mitosis and participates in the spliceosome composition [23]. In yeast, SF3a contains three subunits: Prp21, Prp11, and Prp9. The three SF3a subunits assemble in a stoichiometric ratio of 1:1:1 [21,22]. There is almost no information about Prp11, except that it participates in splicing. As the leading member of U2 snRNP, SF3a splices a series of genes. However, there is little research on it in plants. The spliceosome is the main component of pre-mRNA splicing, and U2 snRNP is one of the five major SnRNPs of the spliceosome. In recent years, there has been more and more research on U2 snRNP, especially SF3b, but less on SF3a. Lorkovic systematically sorted out Arabidopsis U1 and U2 snRNP proteins in 2005, which was based on previous research on the strong conservation of the splicing mechanism between humans and plants [25]. In the SF3a complex, *AT2G32600* of the three members is the gene with the highest homology with animal *SF3a2*. In this study, we performed protein sequence alignment of human SF3a2 and *Arabidopsis* AT2G32600. We defined *AT2G32600* as being the homologous gene of *SF3a2* in *Arabidopsis thaliana* because they have similar sequences. Additionally, it is the only one in *Arabidopsis*. The phylogeny, replication, transcription, and post-transcriptional regulation of the *SF3a2* gene family in plants were studied using combinatorial bioinformatics methods. We chose a total of 83 eligible candidate genes from 61 species for phylogenetic analysis. We analyzed the evolutionary affinity of the homologous genes of *SF3a2* in plants and their similarity and differences in gene structure, protein structure, promoter sequence, and splicing patterns. This will pave the way for future intensive study of *SF3a2* in plants.

## 2. Results

### 2.1. Identification and Phylogenetic Analysis of SF3a2 Genes in Planta

To elucidate *SF3a2* genes in planta, we first searched the amino acid sequence of *Arabidopsis SF3a2* (*AT2G32600*) via BLAST searches in Phytozome v11.0. Then, a total of 83 eligible candidate genes from 61 different species were selected with high matching values, including 51 dicots, 19 monocots, six other land plants, and seven algal species (Appendix A and Figure 1B). We selected the protein sequences of *Arabidopsis* and rice for sequence alignment with humans (Figure 1A). The high similarity of protein sequences indicates that they are homologous genes. It was evident in this table that each species had one to four genes, and 74% of all species had one gene, including *Arabidopsis thaliana* (*AT2G32600*) and *Chenopodium quinoa* (*AUR62026953-RA*) in dicot, *Oryza sativa* (*LOC_Os03g15700.1*) and *Ananas comosus* (*Aco024234.1*) in monocot, *Selaginella moellendorffii* (*180432*) and *Marchantia polymorpha* (*Mapoly0069s0004.1.p*) in other land plants, and *Chlamydomonas reinhardtii* (*Cre06.g252700.t1.2*) and *Micromonas pusillaCCMP1545* (*44451*) in algal species. The percentage that possessed two copies was 20%, including *Arabidopsis helleri* (*Araha.1971s0014.1.p* and *Araha.1905s0002.1.p*) in dicot and *Sphagnum fallax* (*Sphfalx0014s0238.1.p* and *Sphfalx0006s0301.1.p*) in other land plants, while there were none in monocot and algal species; furthermore, *Kalanchoe laxiflora* (*Kalax.1083s0001.1.p*, *Kalax.0420s0030.1.p*, *Kalax.0178s0076.1.p,* and *Kalax.0186s0022.1.p*) and *Zea mays* (*Zm00008a035638_P01*, *Zm00008a001017_P01*, *GRMZM2G100620_P01*, and *GRMZM2G307906_P01*) with the most four genes made up three percent of all species, followed by *Panicum virgatum* (*Pavir.9NG665400.1.p*, *Pavir.9KG512300.1.p* and *Pavir.9KG504400.1.p*) and *Triticum moeaestivum* (*Traes_4DL_87F981684.1*, *Traes_4AS_74660DF99.1* and *Traes_4BL_3EA2CD6B0.1*) that each possessed three copies with the same proportion (Appendix A).

To analyze the phylogenetic organization of the *SF3a2* family, we performed a phylogenetic analysis of 83 different sequences in different species. The phylogenetic tree was constructed based on multiple protein sequences alignment of the 83 genes above via the maximum likelihood (ML) method using RAxML (Figure 1B). The phylogenetic tree was divided into four parts, dicots in blue, monocots in red, other land plants in green, and algal species in yellow; thus, the dicots took up more than half of the tree indicating that most of the genes were of dicots. *Arabidopsis thaliana* marked with a black arrow while *Oryza sativa* represented the monocots (Figure 1B and Appendix A). The location indicated a close relationship. For example, the dicots are clustered together. *Arabidopsis thaliana* was surrounded by *AL4G28000.t1*, *Araha.1971s0014.1.p*, *Carubv10023821m* and *Cagra.0804s0006.1.p*, implying that the five genes had similar evolutionary relationships; *AL4G28000.t1* was much less likely than the other four genes to be in this position with the value of 0.736 because the different bootstrap values represented a difference in credibility. *LOC_Os03g15700.1*, *Brast02G124800.1.p*, *Bradi1g67320.1.p*, *Traes_4DL_87F981684.1*, *Traes_4AS_74660DF99.1*, and *Traes_4BL_3EA2CD6B0.1* were on the same branch; among the rest, *Traes_4DL_87F981684.1* had the value of 1 while *Brast02G124800.1.p* was 0.121 (Figure 1B and Appendix A). In addition, the different genes in the same species were close to each other on the same branch, like with *Triticum moeaestivum, Zea mays*, and *Panicum virgatum*, but with some differences, such as *Kalanchoe laxiflora* whereby four genes existed in different branches. Besides the four species above, the same is true of other digenic species; *Amaranthus hypochondriacus, Glycine max, Gossypium hirsutum, Malus domestica, Physcomitrella patens, Prunus persica,* and *Sphagnum fallax* were the same but, on the contrary, *Arabidopsis halleri, Capsella grandiflora, Capsella rubella, Kalanchoe fedtschenkoi,* and *Populus trichocarpa* were far, suggesting that they were close to others instead of the genes in their species (Figure 1B and Appendix A).

### 2.2. Analysis of Gene Structure and Peptide Domain

The analysis of gene structure and conserved motif compositions provides additional clues about the evolutionary relationships of the *SF3a2* families. With regard to the gene structure, we downloaded the gene structure diagram and cDNA sequences of the 83 sequences from Phytozome v11.0 when searching the genes. In the middle panel (Figure 2), we can see the composition of genes: the number of exons and introns, the UTR presence, and gene length. Their gene structure was different because we searched them according to protein sequence similarity. In the genetic form of other species, we observed that the maximum number of exons was nine while the minimum was one (Figure 2, the middle panel). Seven exons were the most common ones, no matter whether they were dicots, monocots, other land plants, or algal species. Three genes did not have introns, with two (*Araha.1905s0002.1.p* and *Carubv10007569m*) in dicots and one (*13495*) in the algal species, and *Araha.1905s0002.1.p* had no introns, while *Araha.1971s0014.1.p* had six introns as they were in different branches even if they were the same species (Figure 2, the middle panel). The three genes of *Triticum aestivum* were different from each other, either exons or UTR, implying that they were evolutionarily diverse; *Panicum virgatum* was the same. In contrast, the two genes of *Gossypium hirsutum* were about the same in introns and UTR. The intron phase was labeled in the figure (Figure 2, the middle panel) to elucidate the evolutionary relationship, even if we can make it clear how intron splicing and exon splicing will occur in the future.

The Multiple Em for Motif Elicitation (MEME) search tool was subsequently utilized to analyze the conserved motifs in the *SF3a2* families. Ten motifs were illustrated in colored boxes in the right column. By and large, 71 sequences contained all ten motifs with uniform orders. In the dicots, five sequences had nine motifs, with three sequences lacking in Motif 8 and the rest of the two sequences lacking in Motif 1 and Motif 6, respectively; one sequence was lacking two motifs (Motif 9 and Motif 10). In monocots; *Zm00008a001017_P01* was missing Motif 1, Motif 2, and Motif 3, while *Traes_4DL_87F981684.1* just missed Motif 1. Only one sequence was in other land plants; *Sphfalx0006s0301.1.p* was missing two motifs (Motif 9 and Motif 10); one of the three algal species was lacking three motifs (Motif 5, Motif 7, and Motif 10); and the remaining two were lacking in Motif 5 and Motif 1, respectively (Figure 2, the right panel). *Potri.016G097600.1* had one less motif than *Potri.006G121300.1*, suggesting that they went through different evolutionary processes to perform various functions.

As for the peptide level, the same means were with cDNA. All of the sequences possessed the exact domains (Figure 3, the middle panel) named the Zinc-finger of C2H2 type and Pre-mRNA-splicing factor SF3a complex subunit 2 (SF3a2), which directly confirmed that they belonged to the same family. The domains of most sequences were in a similar position except for six sequences, including Kalax.1083s0001.1.p and Kalax.0178s0076.1.p in dicots, and Zm00008a001017_P01, Pavir.9KG512300.1.p, Traes_4DL_87F981684.1, and Aco024234.1 in monocots (Figure 3, the middle panel). Interestingly, ten peptide motifs did not exist in every sequence, even if they had the same domain, SF3a2. The SF3a2 domain contained Motifs 2 and 3 and most areas of Motifs 7 and 5. Only a few sequences had ten motifs covered, Gohir.D09G140400.1.p, Gohir.A09G144600.1.p, Bradi1g67320.1.p, Brast02G124800.1.p, and Sphfalx0014s0238.1.p, while most sequences missed Motif 10 (PPPQGFPGQQM), including all algal species; in addition, Motif 5 (YRVTKQYDPETKQRSLLFQIEYPEIEDNTKPRHRFMSSYEQ) existed in every sequence. Only three sequences missed Motif 1 (MDREWGSKPGSGGAASAQNEAIDRRERLR), two sequences missed Motif 2 (LALETIDLAKDPYFMRNHLGSYECKLCLTLHNNEGNYLAHTQGKRHQTNL), one sequence missed Motif 3 (AKRAAREAKDAPAQPQPNKRK), two sequences missed Motif 4 (VRKTVKIGRPG), one sequence missed Motif 6 (VQPFDKRYQYLLFAAEPYEIIAFKYPSTEIDK), and five sequences missed Motif 7 (TPKFFSHWDPDSKMFTLQLYFKPKPPEAN). In contrast, the sequences that missed Motif 8 (KPPPPPAPNGTGAPGAPPRPP), Motif 9 (PPPPPMANGPRPMPP), and Motif 10 were over 90 percent (Figure 3, the right panel). It is worth mentioning that twelve sequences were reversed between Motifs 8 and 9, including Cucsa.306500.1, Potri.006G121300.1, Potri.016G097600.1, and LOC_Os03g15700. 1 (Figure 3, the right panel). Zm00008a001017_P01 was a minimum of four motifs, followed by Prupe.4G071900.1.p., which had five motifs. In summary, 83 genes in 61 species have a SF3a domain, and they are also similar in gene structure, indicating that SF3a2 is conservative in species evolution.

### 2.3. Promoter Analysis of Tissue, Hormone, and Stress Levels

To obtain hints for how the expression of the *SF3a2* genes may be regulated, potential cis-elements in the 1500 bp promoter regions upstream of the genomic sequence of the *SF3a2* genes were identified by searching the PLACE database. As a result, multitudinous cis-elements of tissue, hormone, and stress were found to demonstrate the differences in gene regulation (Appendix A).

In tissue expression regulation, twelve cis-elements existed to regulate their expression of organizational differences including the RY-element (cis-acting regulatory element involved in seed-specific regulation), as-2-box (involved in shoot-specific expression and light responsiveness), as1 (cis-acting regulatory element involved in the root-specific regulation), GCN4_motif (cis-regulatory element involved in endosperm expression), HD-Zip 2 (element involved in the control of leaf morphology development), HD-Zip 1 (element involved in differentiation of the palisade mesophyll cells), Skn-1_motif (cis-acting regulatory element required for endosperm expression), CAT-box (cis-acting regulatory element related to meristem expression), and four cis-acting regulatory elements related to meristem specific activation named CCGTCC-box, dOCT, NON-box, and OCT (Figure 4). Thus, seventy-seven sequences possessed the Skn-1_motif indicating that most of the genes played an essential role in endosperm expression, and that *SF3a2* family genes played a small part in root-specific regulation and seed-specific regulation because only two sequences had as1 and four sequences had RY-element. HD-Zip 1 and HD-Zip 2 coexisted at *Kalax.1083s0001.1*, *Cucsa.306500.1*, and *Glyma.15G005900.1* in the exact same positioning, suggesting that they are functionally similar. *LOC_Os03g15700.1* had five cis-acting regulatory elements, while five sequences had none (Figure 4), including *Thhalv10017051m*, *Migut.J00456.1*, *Manes.09G061900.1*, *Zm00008a035638_T01*, and *Traes_4DL_87F981684.2*, indicating that they played a different part in tissue-specific gene expression.

As for hormone and stress levels, twenty-seven cis-acting regulatory elements are shown in Figure 5. Some cis-elements relating to auxin responsiveness contained AuxRE, AuxRR-core, TGA-element, and WUN-motif, some related to abscisic acid responsiveness contained ABRE, CE3, and motif IIb, and some pertaining to low-temperature responsiveness contained DRE, C-repeat/DRE, LTR, and MBS (Figure 5). Different sequences possessed different elements indicating that different *SF3a2* family genes participated in other regulatory mechanisms. ARE, CGTCA-motif, MBS, TC-rich repeats, TCA-element, and TGACG-motif existed in most sequences, while AuxRE, DRE, CE3, ELI-box3, SARE, and motif IIb were in fewer sequences. *AT2G32600.1* had four elements named ARE, CGTCA-motif, GARE-motif, and TGACG-motif, *LOC_Os03g15700.1* also had four elements named ARE, CGTCA-motif, LTR, and TGACG-motif (Figure 5); this gives support to the idea that the two genes had the exact same mechanism in anaerobic induction and anaerobic induction, but different mechanisms in gibberellin responsiveness and low temperature responsiveness.

### 2.4. Expression Pattern for Tissue and Hormone

Since expression profiling was a valuable tool for understanding gene function; we summarized several representative *SF3a2* genes using microarray data from eFP browser (Figure 6). Four plant species have multiple *SF3a2* genes: the two soybean genes performed the same tissue expression pattern; this was higher in the root tip and lower in the nodule with the standard line of root, even if their discrepancy in the promoter cis-acting regulatory element was huge, and the RY-element existed in *Glyma.15G005900.1* but not in *Glyma.13G367400.1* (Figure 4 and Figure 6A). As for single-gene species, in dicots, *Arabidopsis* was highly expressed in dry seed and shoot apex, but had low expression in the leaf (Appendix A). Singled out the seed data; SD2 and the embryo globular stage were two periods with the highest expression, while the seed coat mature green stage was lower; for individual shoot data, the expression quantity was generally lower than ATML1 but was the lowest in CLV3 (Appendix A); and for cottonwood (*Potri.006G121300.1*) and tomato, the expression levels of each group were usually lower than their respective roots (Appendix A). In monocots, we analyzed purple false brome and rice. Purple false brome contained a generally high expression compared to root, especially in the internode (60 days). However, because of a shoot-related element named as-2-box, there was no especially prominent high expression situation in rice, but the expression was particularly low in young leaves (Appendix A).

Gene expression analysis revealed the changes in transcript abundance in response to hormones. Data on BAR showed that IAA treatment can improve the expression level of *SF3a2*, and ABA treatment can reduce the expression level of *SF3a2* (Figure 6B). However, in our experiment, IAA and ABA treatment were shown to dramatically increase the expression level of *SF3a2* (Figure 6C) since it has no cis-action elements related to IAA and ABA, but our experiment showed that *SF3a2* can correspond to IAA and ABA. Given that *Arabidopsis* had a cis-acting regulatory element critical to the anaerobic induction, we neatened the information after oxidative treatment and hypoxia stress treatment (Figure 6D,E). As a consequence, the expression level in the experimental group of the shoot increased first and then decreased obviously following the control group, and the change in root was inconspicuous both in the experimental and control group when given oxidative treatment; hypoxic treatment is more straightforward than oxidative treatment, and the expression level was improved not only in total RNA but also in polysomal RNA, although total RNA was more prominent. Various expression patterns may be related to the cis-acting regulatory elements in the promoter region. SF3a2 was localized in the nucleus (Figure 6F and Appendix A).

### 2.5. Analysis of Protein–Protein Interaction and 3D Structure Conservation Analysis

To find out the working principle, the search tool STRING (https://string-db.org/) (25 February 2023) was used to perform the protein interaction networks of *Arabidopsis thaliana* (AT2G32600) and *Oryza sativa* (LOC_Os03g15700.1) with their amino acid sequence. Then, ten predicted interaction proteins of SF3a2 protein were obtained, including AT1G09760 (U2A), AT1G09770 (CDC5), AT3G19840 (PRP40C), AT4G21110 (AT4G21110), AT5G06160 (ATO), AT5G64270 (AT5G64270), AT1G31870 (AT1G31870), AT3G13200 (EMB2769), AT1G77180 (SKIP), and AT5G27720 (emb1644) in Arabidopsis (Figure 7A); thus, U2A, CDC5, PRP40C, ATO, SKIP, AT5G64270, AT1G31870, EMB2769, and emb1644 participated in the processing of mRNA splicing, suggesting that AT2G32600 belonged to the spliceosome components. AT4G21110 was a G10 family protein gene.

There were also ten interaction proteins in rice (Figure 7B), including OS02T0234800-01 (OsJ_06008), OS02T0827300-01 (OsJ_08964), OS03T0717600-01 (OsJ_12358), OS04T0348300-01 (OS04T0348300-01), OS10T0466300-01 (PRP19), OS10T0485000-01 (OS10T0485000-01), OS03T0339100-01 (OS03T0339100-01), OS08T0178300-01 (OS08T0178300-01), OS09T0249600-01 (OS09T0249600-01), and OS06T0608300-01 (OS06T0608300-01). Thus, OsJ_06008, OsJ_08964, OsJ_12358, PRP19, Os03g0339100, and OS09T0249600-01 participated in the processing of mRNA splicing. OS04T0348300-01 was a putative uncharacterized protein. OS10T0485000-01 was an FF domain-containing protein. OS06T0608300-01 was a putative uncharacterized protein. Above all, *SF3a2* family genes and their functional partners were involved in RNA splicing and some other biological processes such as growth and development.

*AT2G32600*, for example, had a 3D model reconstructed to analyze the conservation of the gene (Figure 7C). There were variable regions in the structure, suggesting that these regions had a higher probability of variable splicing, while some regions were very conservative and did not change. The conservative coefficients were divided into nine levels. The higher the value, the more conservative it was. In addition to the protein subtypes, three-dimensional models were reconstructed. Due to incomplete template structure (PDB ID: 5MQF), only part of residues (1–210) of *AT2G32600* were constructed as being homologous structures. In *AT2G32600*, the hit range of the Hmmer model starts at 1 and ends at 210. Therefore, homologous structures are sufficient to represent SF3a2 splicing factors in plant species. In the aligned sequence, ConSurf Gradehad more than five amino acid residues. According to the model (Figure 7C), there are a few charged polar amino acids, such as Arg3 and Arg27, which means that electrostatic interaction is required for binding with RNA. However, from sequence alignment, the low ConSurf Grade is caused by the mutation of individual species. To sum up, the core area of SF3a2 is highly conservative.

### 2.6. Genomic Synteny Relationship of SF3a2 Genes

Hom0109y and Synteny were analyzed on the whole genome scale, the phylogenetic tree of the genome was constructed, and evolutionary differences among *SF3a2* genes were analyzed (Figure 8). Forty-nine species were used to build this phylogenetic tree via the data in Piece 2. They had distinct differences in exon evolution even though they belonged to the same family. The number of exons was between one and seven. None of these exons were wholly conserved in 45 species. The five exons between *AT2G32600.1* and *Carubv10023821m* were matched while the next gene *Cagra.4714s0005.1* had one exon matched; the other exon was a mismatch, and the two aligned exons shared the same frame. The same occurred with *Arabidopsis*: *LOC_Os03g15700.1*, *Sobic.001G428400.1*, and *Bradi1g67320.1* were precisely matched between the same middle five exons. The central five exons were more evolutionarily conservative than the other two exons in 45 species of *SF3a2* genes. The SF3a2 domain contained the complete fourth and fifth exons, while the third and sixth exons were in the sequence except for a few bases. There was just one mismatch, and the two aligned exons shared the same frame between *Pp3c9_13030V3.1* and *180432*, while there was no connection between *44451* and *63373* (Figure 8). In conclusion, the exons of different genes were evolutionarily correlated and uncorrelated, indicating that they had other functions.

## 3. Discussion

Our results show research into the family evolution of the homologous genes of *SF3a2*, a member of the spliceosome, in plants. We studied the gene structure, protein structure, and conservative motif analysis of *SF3a2* in plants and analyzed their promoter regions. Meanwhile, we analyzed their spatial and temporal expressions and predicted the interaction proteins of several genes. Furthermore, we examined the coevolutionary relationships among some species, indicating the evolutionary relationship between exons. Our research lays a foundation for studying alternative splicing of the spliceosome in plants.

### 3.1. The Evolutionary Correlation of SF3a2s Suggests That They May Have Similarities in Function

As a U2 snRNP core component, SF3a participates in pre-mRNA splicing and regulates cell morphology, growth, development, and differentiation [26]. *SF3a2* exists in a few species of animals as one of the three subunits of SF3a, in humans (*SF3a66*, *SF3a2*, and *SAP62*), in yeast (*Prp11* and *SAP62*), and in mice (*SF3a2* and *SAP62*) [27,28,29]. In addition to interacting with *SF3a3*, *SF3a2* binds to 15s U2 snRNP through a zinc finger structure. The C-terminal of *SF3a2* has more repeat GVHPPAP than the Prp11 of yeast, which indicates that *SF3a2* has other functions besides binding to U2 snRNP [30]. Now, according to protein sequence similarity, we researched SF3a2s in plants and clarified some of their relationships. In *Arabidopsis*, *ATO* was defined as the homologous gene of *SF3a3*, PPI suggested that AT2G32600 can interact with ATO, and the *SF3a2* genes were conservative (Figure 7C). Spliceosome contents were localized in the nucleus to perform functions, and SF3a2 was localized in the nucleus (Figure 6F). This indicates that ATSF3a2 participates in RNA splicing, like with SF3a2 in the mammal.

Eighty-three eligible candidate genes from 61 different species were selected with a high matching value to construct the evolutionary tree (Appendix A and Figure 1B). In the evolutionary relationship, they are relatively clustered, with monocotyledons being clustered and dicotyledons being clustered. This suggests a relatively distant relationship. For example, three genes of *Arabidopsis* were clinging together to indicate a closer kinship. Their protein structure was almost in agreement with two domains: Zine-finger and SF3a complex subunit 2 (Figure 3). The gene structure was very similar, including exons and introns in multiple genes (Figure 2). This indicates their homology and functional similarity. Each of the genes contains a tissue-specific cis-element named Skn-1_motif, suggesting that they have a similar function in endosperm development (Figure 4). As for the characteristics of the hormone- and stress-related cis-elements, they may participate in some abiotic stresses such as abscisic acid (ABA), gibberellin (GA), temperature, and oxygen response (Figure 5). The results of collinearity analysis showed that the exons of most genes were closely related in evolution (Figure 8). Plants have formed mechanisms to protect critical genes throughout the evolution process; the above results show that *SF3a2s* may have functional similarity in planta.

### 3.2. Differences between Species Indicate the Functional Differentiation of SF3a2s in Plants

The current plant evolution hypothesis believes that higher plants have experienced multiple whole genome duplication (WGD) events. In each WGD event, the footprint of gene family expansion implies gene replication, providing a cornerstone for functional diversification [31]. SF3a2, as the core member of the spliceosome, should be widely found in most species. However, with the evolution of plants, doubling events, deletion events, and mutation events may produce differences leading to functional differences.

Two genes were at a distance in the same species, such as *Carubv10007569m* and *Carubv10023821m*. In addition, their genetic structure varied greatly; *Carubv10023821m* had seven exons, while *Carubv10007569m* had no exons (Figure 2). These genes not only differed in the motif of a gene but also differed in the motif of a peptide (Figure 2 and Figure 3). CCGTCC-box exited in monocotyledon, lower plants, and alga, while it rarely appeared in the dicotyledon (Figure 4). The expression patterns of tissues in different species were different (Appendix A). The response of different genes to abiotic stress is also different (Figure 5 and Figure 6). A few genes did not appear in the collinearity analysis, suggesting that they had closer kinship but were unrelated to exons (Figure 8). These results indicated the functional differentiation of *SF3a2s* in plants.

Plants are very diverse, whether this is related to the petal color of angiosperms, the leaf shape of ferns, the branching pattern of gametophytes in mosses, or their interaction with microorganisms and the environment [32,33,34]. In the process of plant evolution, some genes will be lost, and some new genes will be generated to adapt to environmental changes. The history of gene duplications in plant evolution has led to some large polygenic families. A very prominent feature of the *Arabidopsis* genome is the numerous MYB transcription factors [28]. The functions of members of the same family are similar. When a gene is deleted, members of the same family can partially supplement the functions of the gene, so that the phenotype can only appear when several genes are knocked out. SF3a, as the core content of U2 snRNP, is relatively conserved in the process of evolution. Our results also indicate the conserved form of *SF3a2s*, from alga to monocotyledons. The spliceosome is involved in splicing a series of genes, which are divided into two categories: constitutive splicing and alternative splicing. In recent years, with the extensive research into alternative splicing, we have gained a new perspective on gene-to-protein regulation. The members of the spliceosome are genes with alternative splicing. Are there any conditions for the recruitment of different transcripts by the spliceosome? How will the recruitment of different transcripts affect the splicing of downstream genes? What impact will different transcripts produced by the same gene have on plant growth, development, and plants’ biotic and abiotic stresses? Here, we have listed the existence of alternative splicing transcripts in *SF3a2s* in different taxonomic species (Appendix A). We propose this point here, and more in-depth experiments and functional analysis are needed to verify it in the future.

## 4. Materials and Methods

### 4.1. Identification and Phylogenetic Analysis of SF3a2 Genes in Planta

We used the protein sequence of *Arabidopsis thaliana* (*AT2G32600*) in Phytozome v11.0 (https://phytozome.jgi.doe.gov/pz/portal.html) (1 June 2018) to make searches, and we used BLASTp program to search in Phytozome v11.0 for similar (e-value cutoff = 1 × 10^−10^) sequences in all known plant species [35]. The resulting protein sequences were detected by the PF08231 model of HMMER [36]. Then, a total of 83 eligible candidate genes from 61 species were selected with a high matching value. Each species had no more than four genes. We summarized the species, definite gene name, and the gene number of every species in detail in Appendix A. These sequences form the phylogenetic tree in the next section.

### 4.2. Phylogenetic Analysis of SF3a2 Gene Family in Planta

These sequences were used to construct an evolutionary tree to determine the classification and evolutionary relationship among the plant *SF3a2* family genes. We used PhyML v3.0 to construct a maximum likelihood (ML) tree using the JTT+G model [37], and then used FigTree v1.4.3 to represent and edit the tree [38].

### 4.3. Analysis of Gene Structure and Peptide Domain

We used the protein sequence of AT2G32600 to explode the target proteome of each species, select a numerical sequence, obtain experimental data, and obtain a large number of candidate genes. The gene structure was downloaded from Phytozome v11.0. The intron phase was displayed in the Gene Structure Display Server 2.0 (http://gsds.cbi.pku.edu.cn/) (1 July 2018). Protein domain prediction was conducted via biosequence analysis using profile hidden Markov models: HMMER (https://www.ebi.ac.uk/Tools/hmmer/) (1 July 2018). The MEME analysis (http://meme-suite.org/tools/meme) (19 July 2018) was used to obtain their conserved motifs of cDNA and peptides. We searched their cDNA and peptide sequences, imported all of the sequences to the Multiple Em for Motif Elicitation website, selected 10 motifs, and then obtained the conserved motifs of every sequence.

### 4.4. Promoter Analysis of Tissue, Hormone, and Stress Levels

The 1500 bp in front of every gene was selected as the promoter sequence from Phytozome v11.0. The putative sites with tissue-specific expression, hormones, and stress responses were analyzed using PlantCARE (http://bioinformatics.psb.ugent.be/webtools/plantcare/html/) (1 August 2018), and their positions were marked on the 5′-flanking sequence.

### 4.5. Microarray-Based Expression Analysis

The website of the BAR (http://www.bar.utoronto.ca/) (1 September 2018) and its data analysis tools for plant biology were used to find the genes that previous research had certified as being the expression of different tissues at different times. The developmental map and hormones were chosen to find out the expression quantity. The BAR HeatMapper Plus Tool (http://bar.utoronto.ca/ntools/cgi-bin/ntools_heatmapper_plus.cgi) (27 September 2018) was used to draw a map.

### 4.6. Plant Materials and Growth Conditions

*Arabidopsis thaliana* ecotype Columbia-0 (Col-0) was used for all experiments in this study. In the aseptic workbench, the seeds were first sterilized with 70% ethanol, sterilized with 20% bleach for 20 min, then washed with deionized water 5 times, and then the seeds were spread on the Murashige and Skoog solid medium. They were transferred to the light chamber after 4 °C dark treatment for 2 days. The seedlings growing for 10 days were treated with different hormones (50 μM) for 4 h, and RNA was extracted for quantitative analysis. Other tissue RNA was extracted for quantitative analysis, such as root, stem, leaf, flower, silique, seed, seedling, and shoot apex. qPCR was performed using the CFX96 Touch Real-Time PCR Detection System (Bio-Rad, Hercules, CA, USA) with SYBR Premix ExTaqII (Takara, Beijing, China). *GAPDH* was used as a reference in qPCR analysis. Data are shown as the mean expression ± SD. The primers were SF3a2-qRT-F (CATCTTGGAAGCTATGAGTGTAAAC) and SF3a2-qRT-R (CTACCAATTTTAACTGTTCTGCG).

### 4.7. Analysis of Protein–Protein Interaction

The search tool STRING (https://string-db.org/) (25 February 2023) was used to create the protein interaction networks of *Arabidopsis thaliana* (*AT2G32600*) and *Oryza sativa* (*LOC_Os03g15700.1*) with their amino acid sequences. To build the protein interaction networks, we chose ten genes from known interactions (from curated databases; they were experimentally determined).

### 4.8. Three-Dimensional Protein Analysis

Based on the known human spliceosome-like protein structure activated via splicing in Step 2 (PDBID: 5MQF, ID: 36%, E-value: 3 × 10^−27^, query coverage: 70%), the Swiss model server [39] was used to model the homology of *AT2G32600*, using the ML method to calculate the amino acid preservation score through the ConSurf Web server [40]. The multiple sequence alignment, the tree files, and the homologous structure were provided as input attributes. Models and figures were drawn using Pymol [41].

### 4.9. Vector Construction and Subcellular Localization

The full-length coding sequence (CDS) of *ATSF3a2* was subcloned into the pMDC43 vector via gateway technology (Invitrogen, Carlsbad, CA, USA) using specific primers (F:GATGAACTATACAAAGGCGCGCCGATGGATAGAGAATGGGGTTCCA and R:GGCGGCCGCTCTAGAACTAGTTCACATGTTAGGATGTCCGAAACC). Then, the constructed *Agrobacterium* tumefaciens transiently transformed tobacco cells (*Nicotiana benthamiana*). The fluorescence signal of green fluorescent protein (GFP) was detected using an LSM 880 laser scanning confocal microscope (Carl Zeiss, Batenwerburg, Germany) with the excitation and emission wavelengths set to 488 nm and 505–550 nm, respectively.

### 4.10. Analysis of Genomic Synteny Relationship

The analysis tool of Piece 2 (http://www.bioinfogenome.net/piece/search.php) (1 December 2018) was used to study the genomic synteny relationship of *Arabidopsis thaliana* (*AT2G32600*) with its gene ID. We eliminated 5 genes that did not exist in the 49 species.

## Figures and Tables

**Figure 1 ijms-24-05232-f001:**
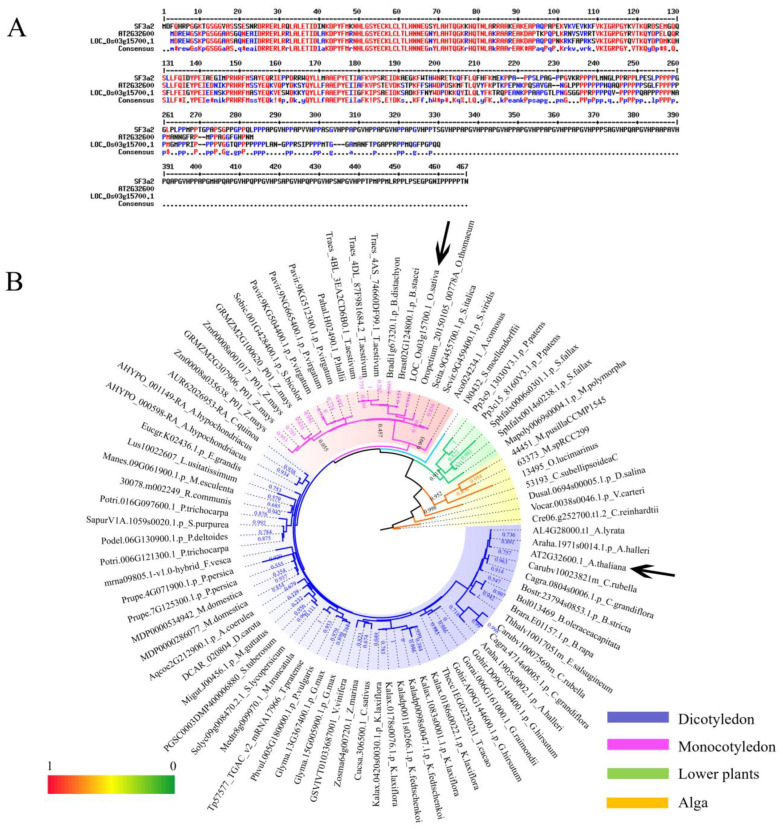
Phylogenetic analysis of the *SF3a2* genes in the plant lineage. (**A**) Sequence alignment of human *SF3a2*, *Arabidopsis AT2G32600*, and rice *LOC_Os03g15700.1*. (**B**) Phylogenetic analysis of *SF3a* gene family using maximum likelihood methods as implemented in software PhyML v3.0. The evolutionary tree consists of 83 genes selected in this research. Different colors represent different floristic compositions, as marked at the bottom right side of the picture. Bootstrap values are marked at the branches. The assigned genes of *Arabidopsis thaliana* and *Oryza sativa* are pointed out by black arrows. The Latin abbreviations and the specific names of every gene are listed in Appendix A.

**Figure 2 ijms-24-05232-f002:**
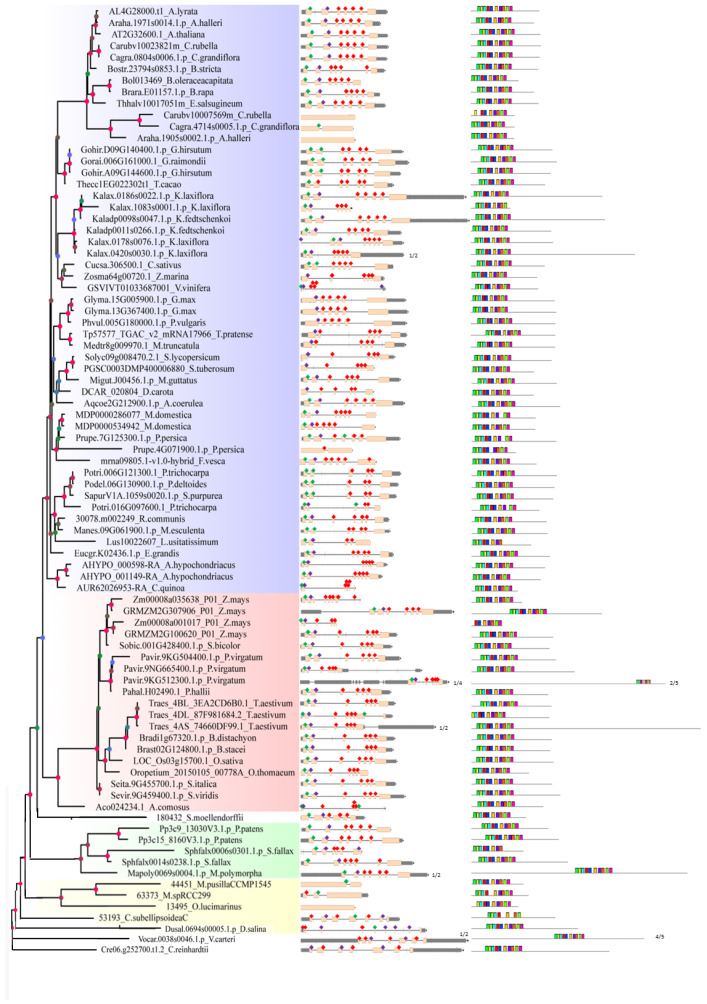
Gene structure comparisons and multiple Em for Motif Elicitation (MEME) analysis of plant *SF3a2* genes. The phylogenetic relationship, gene structure, and conserved motifs of cDNA sequences are listed on the picture’s left, middle, and right sides. Some long genes were curated to fit this picture. The scale is labeled at the end of every image. The conserved sequences are listed below the picture.

**Figure 3 ijms-24-05232-f003:**
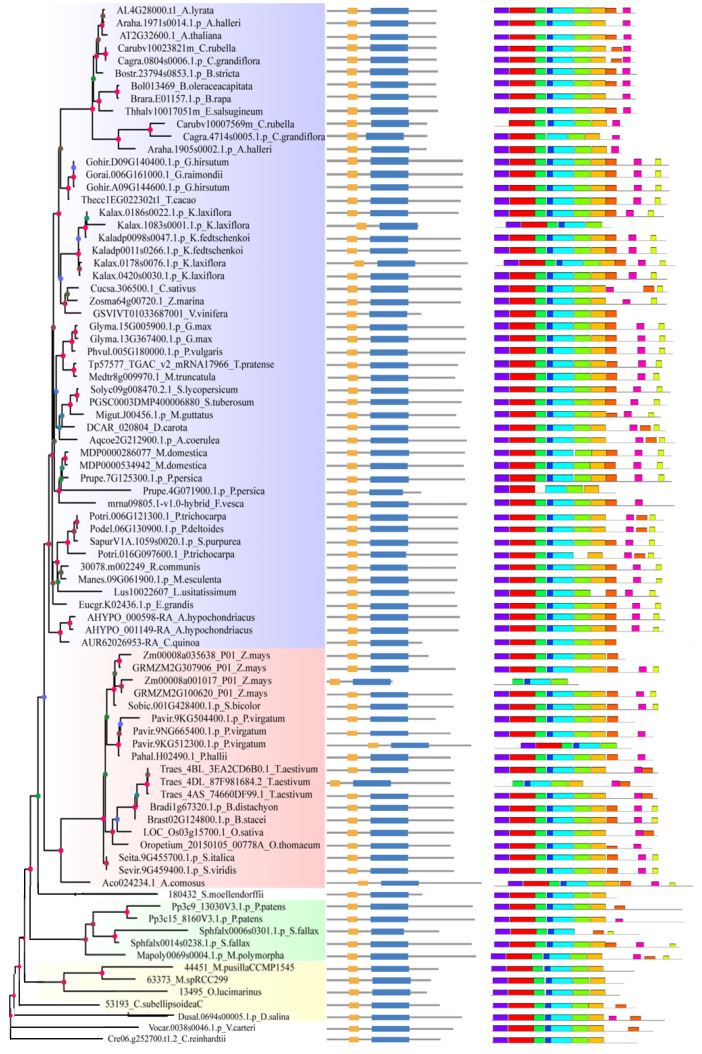
Protein domain and multiple Em for Motif Elicitation (MEME) analysis of plant SF3a2 proteins. The phylogenetic relationship, gene structure, and conserved motifs of cDNA sequences are listed on the picture’s left, middle, and right sides. The conserved sequences are listed below the image. The main protein domains are labeled below the picture.

**Figure 4 ijms-24-05232-f004:**
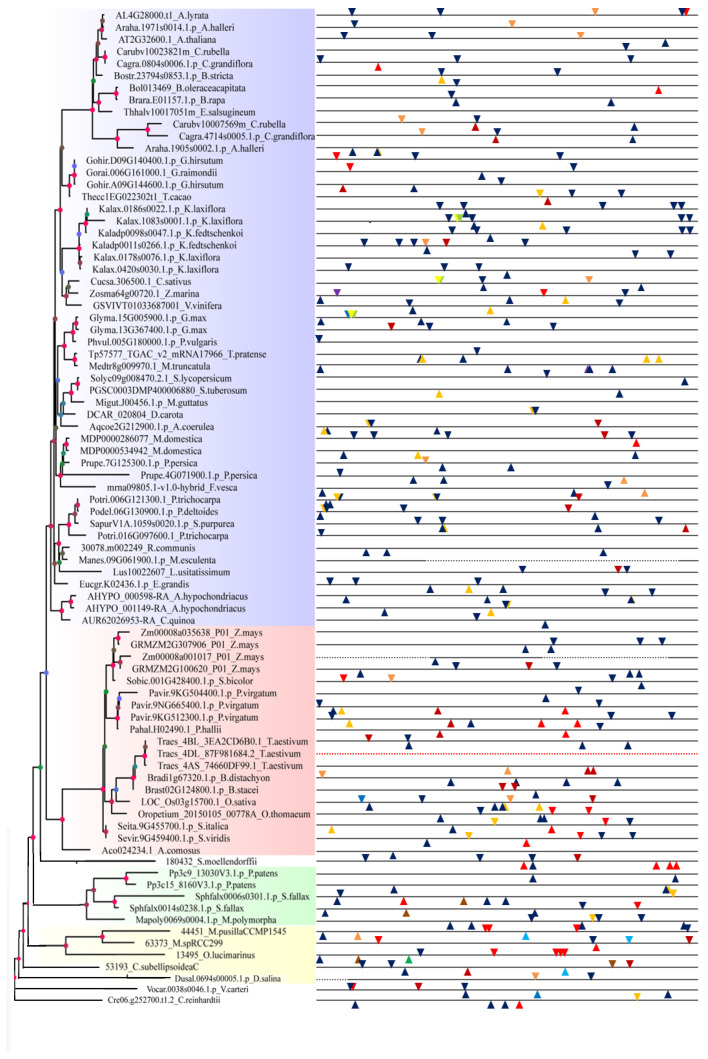
Analysis of tissue-specific *cis*-element in the 5′-flanking sequences of plant *SF3a2* genes. Twelve putative *cis*-elements are represented by different symbols as indicated in different colors. The function of every motif is listed next to the symbol. These motifs are marked along 1.5kb 5′-flanking sequences before every gene. A downward triangle means that this element is on the positive-sense strand. On the contrary, the upturned triangle means it is on the antisense strand. The red dotted line shows no sequences. The black dotted line represents the region of annexed base N. The full line represents the regions with explicit basic pairs.

**Figure 5 ijms-24-05232-f005:**
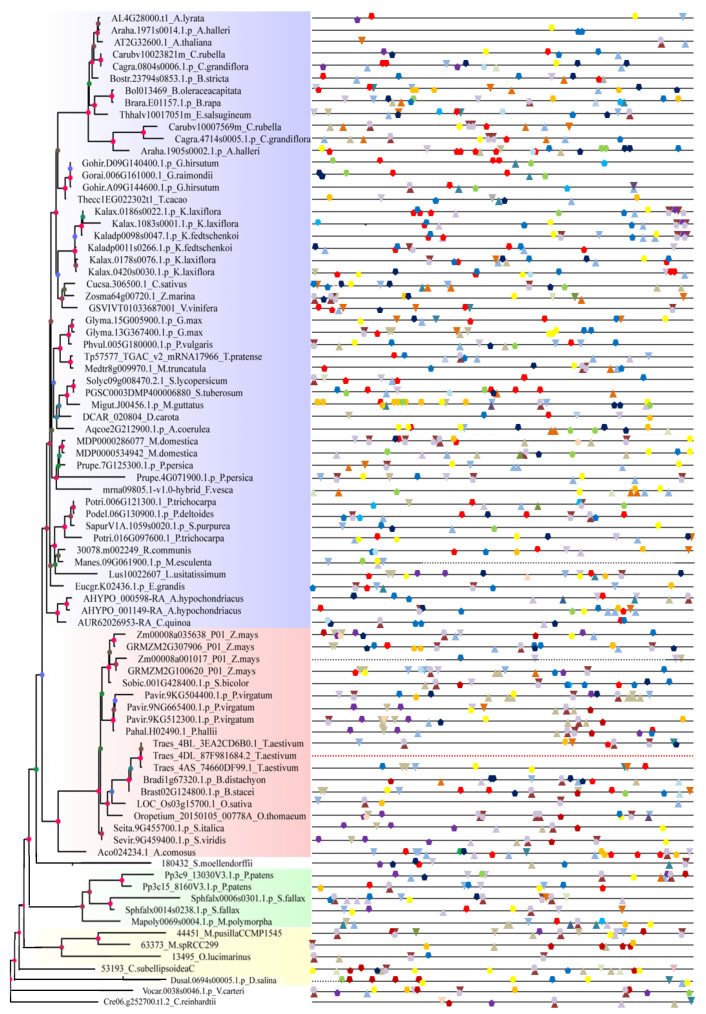
Analysis of hormone- and stress-related cis-element in the 5′-flanking sequences of plant *SF3a2* genes. Different symbols in different colors represent a total of 27 putative cis-elements. The function of every motif is listed next to the symbol. These motifs are marked along 1.5kb 5′-flanking sequences before every gene. Downward triangles and pentagons mean that this element was on the positive-sense strand. On the contrary, the upturned triangles and pentagons mean that they were on the antisense strand. The red dotted line shows no sequences. The black dotted line represents the regions of annexed base N. The full line represents the regions with explicit basic pairs.

**Figure 6 ijms-24-05232-f006:**
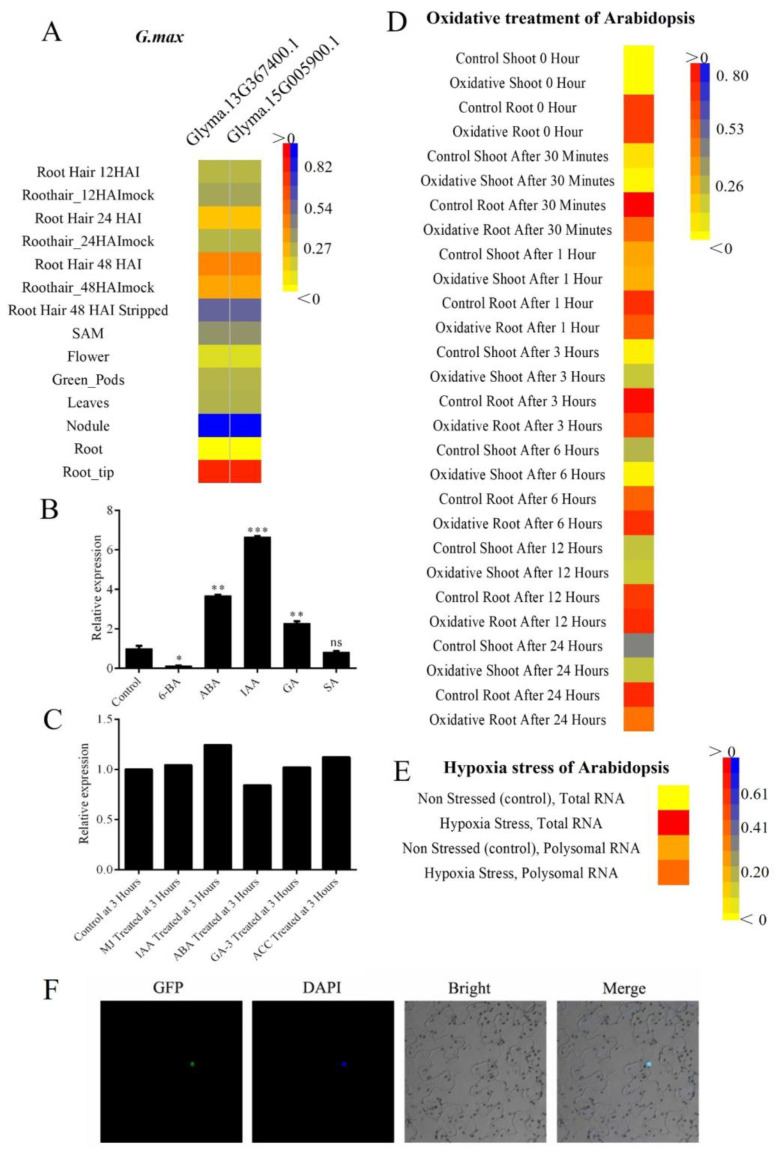
Spatial and temporal expressions of *Glycine max SF3a2* and *Arabidopsis thaliana SF3a2*. (**A**) Spatiotemporal expression patterns of two soybean genes. (**B**) Expression of *ATSF3a2* under different hormone treatments with qPCR. *GAPDH* is the internal reference. The results shown are representative of three independent experiments. *** *p* < 0.001; ** *p* < 0.01; * *p* < 0.05; ns: no significant difference (Student’s *t* test). (**C**) Spatiotemporal expression patterns of *ATSF3a2* under different hormone treatments. Data from BAR. (**D**) Spatiotemporal expression patterns of *ATSF3a2* under oxidative treatments. (**E**) Spatiotemporal expression patterns of *ATSF3a2* under hypoxia treatments. Expression data were extracted using the plant eGFP browser and are presented in heatmap format. Red and blue represent high and low expression in different tissues and stages, respectively. The bigger the difference, the darker the color. (**F**) Nuclear localization of ATSF3a2 in tobacco cells. Nuclei were labeled with 4′,6 -diamidino-2-phenylindole (DAPI).

**Figure 7 ijms-24-05232-f007:**
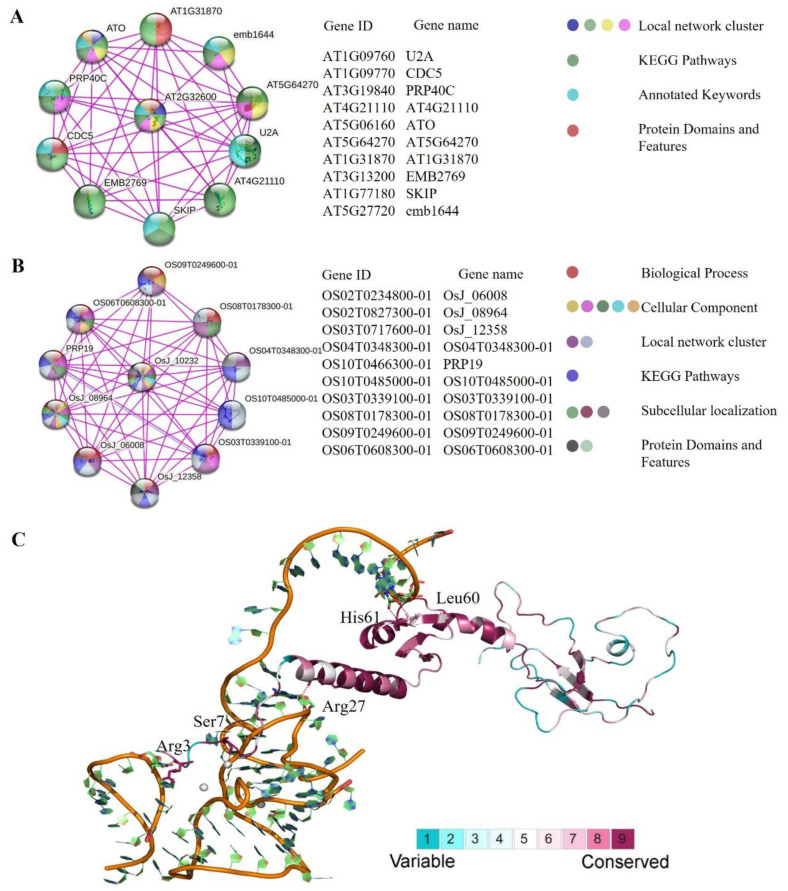
Protein–protein interactions (PPI) with SF3a2. (**A**) Proteins interacting with ATSF3a2. (**B**) Proteins interacting with OsSF3a2. Different spheres represent different genes. The different colors in the sphere indicate the basis of their interaction, listed on the right. The gene ID and gene name are listed in the middle. (**C**) Ribbon representations of plant SF3a2s. The 3D structure representation of plant SF3a2s is colored according to the degree of conservation among the selected 83 SF3a2s sequences based on the homology structure of Arabidopsis. The color bar from blue to red indicates the increase in conservation. The key residues are labeled according to the ATSF3a2s sequence AT2G32600.

**Figure 8 ijms-24-05232-f008:**
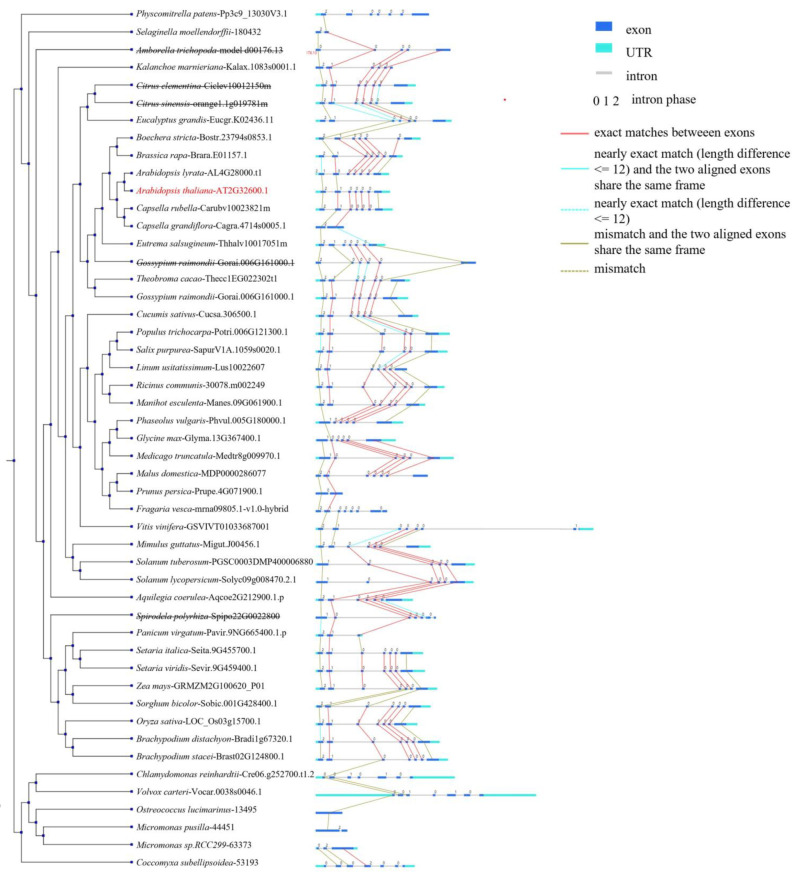
Collinearity of the *SF3a2* genomic blocks in representative angiosperm genomes. Collinearity analysis results between plant *SF3a2s* genomic blocks. The species and gene ID are listed in the middle. Blue squares represent CDS, light blue squares represent UTR, gray lines represent introns, and the number at the top indicates the intron phase. The collinearity between different exons is marked by lines of different colors, as shown on the right.

## Data Availability

Not applicable.

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
