# Peer review of "Phylogenetic Analysis of Spliceosome SF3a2 in Different Plant Species"

_ijms, 2023, doi:10.3390/ijms24065232_

Round 1

Reviewer 1 Report

The paper titled: “Evolution and functional analysis of spliceosome SF3a2 in different plant species” is an interesting manuscript because it addresses the topic of evolutionary relationship in plants with an interesting molecular approach, innovative and very detailed.

I recommend minor revision

I suggest to the authors to move part of the data illustrated in the supplementary materials into the main paper. This facilitates the reading of the paper even for those less experienced in this topic.

However, the manuscript is well structured and the results are well illustrated and merits to be published on IJMS.

Reviewer 2 Report

Tian, Das, et al investigated the phylogenetic distribution of the U2 snRNP splicing factor SF3A2 in different plant species. The authors found that the gene is present in the different species investigated but with diverse features (number of copies, number of exons/introns, transcription factors binding sites…). In addition, the authors show that the expression of SF3A2 could be tissue-regulated and hormone/stress-regulated.  

The paper could be of interest but I currently have several comments.

I am not a phylogeny specialist so I will not comment on this part of the manuscript.

Comments:

1.       It remains unclear why the authors focused specifically on SF3A2, why not the SF3A complex or other SF3A subunits?

2.       I do not understand the scientific purposes of showing the DNA (Figure 2) and peptide (Figure 3) motifs enrichment. The authors do not expand on the motifs in the text so it is unclear what the readers is supposed to get from these enriched motifs.

3.       Figure 6B-C: From my understanding, these are experiments performed by the authors, which should be stated more clearly in the text. In addition, the number of biological replicates needs to be provided (looks line there is only one replicate in Figure 6C) and statistical tests should be performed. The associated Methods section provides only the SF3A2 primers so it is unclear if there is a normalization to a housekeeping gene (if yes, sequence has to be provided and more details needs to be written in the Methods section).  

4.       Figure 6F: While the U2 snRNP is known to be located in the nucleus, I do not think the authors can conclude much from a single spot in their experiment. Clearly, this will need some more convincing results.

Reviewer 3 Report

The manuscript describes the identification and evolutionary analyses of the splicing factor 3a subunit 2. Although the title of the manuscript includes "functional analysis," little experimental evidence for the physiological function of SF3a2 in plants has been presented. In silico analysis has shown the conservation of motifs, introns, and cis-elements. How these are related to the SF3a2 function should be investigated. In addition, the manuscript has the following concerns. Thus, the reviewer cannot recommend acceptance of the manuscript.

1) It is unclear why the manuscript focuses on SF3a2 among the splicing complexes. It is also unclear why At2g32600 was chosen as SF3a2 in Arabidopsis. Are there any other Arabidopsis proteins that show homology to human SF3a2? These points should be clearly stated in the Introduction or start of the Results.

2) The description of the Methods and Results for qRT-PCR is unclear. Figures 6B and 6C treat overlapping phytohormones, but the effects of IAA and ABA are different. What is the difference between Figures 6B and 6C? Materials and Methods state that each phytohormone was treated for 4 hours (line 364), but Figure 6C describes 3 hours. Materials and Methods only state primers to amplify SF3a2 (lines 366-367). qRT-PCR results should be normalized using expression levels of endogenous control genes such as actin, ubiquitin, or 18S rDNA.

3) The manuscript states that the analysis of protein-protein interactions using the STRING database shows only those for which there is experimental evidence (lines 372-373). However, the reviewer found an interaction between AtSF3a2 (At2g32600) and At5g64270 is based on the homologs in other species in the STRING database. The manuscript should carefully describe what evidence each interaction is based on. Experimental evidence using Arabidopsis or rice proteins is required for interactions that are based on information from other species homologs.

Description in STRING for the interaction between AtSF3a2 (At2g32600) and At5g64270

Experimental/Biochemical Data:     none, but putative homologs were found interacting in other organisms (score 0.985).       

Association in Curated Databases:   none, but putative homologs are reported to interact in other organisms (score 0.793).

Round 2

Reviewer 2 Report

The authors have answered my comments.

Author Response

Thank you for your valuable comments. Because of your suggestions, the revised article will become better, and readers can get more valuable information. Thank you again for your help.

Reviewer 3 Report

The revised manuscript has adequately addressed the issues raised in the previous review. The manuscript is acceptable for publication.

Author Response

(The authors gave the same response as above.)
